# Whole-Transcriptome Sequencing Reveals Characteristics of Cancer Microbiome in Korean Patients with GI Tract Cancer: *Fusobacterium nucleatum* as a Therapeutic Target

**DOI:** 10.3390/microorganisms10101896

**Published:** 2022-09-23

**Authors:** Hyeok Ahn, Kyungchan Min, Eulgi Lee, Hyun Kim, Sujeong Kim, Yunjae Kim, Gihyeon Kim, Beomki Cho, Chanyeong Jeong, Yeongmin Kim, Hansoo Park

**Affiliations:** Department of Biomedical Science and Engineering, Gwangju Institute of Science and Technology (GIST), Gwangju 61005, Korea

**Keywords:** gastrointestinal tract cancer, whole-transcriptome sequencing, G2M checkpoint, cancer microbiome, *Fusobacterium nucleatum*

## Abstract

Remarkable progress has occurred over the past two decades in identifying microbiomes affecting the human body in numerous ways. The microbiome is linked to gastrointestinal (GI) tract cancer. The purpose of this study was to determine if there is a common microbiome among GI tract cancers and how the microbiome affects the disease. To ensure ethnic consistency, Korean patients with GI tract cancer were selected. *Fusobacterium nucleatum* is an enriched bacteria in all cancer tissues. *F. nucleatum* is a Gram-negative obligate anaerobe that promotes colorectal cancer. Through Gene Set Enrichment Analysis (GSEA) and Differentially Expressed Genes (DEG) analyses, the upregulation of the G2M checkpoint pathway was identified in the *F. nucleatum*-high group. Cell viability and G2M checkpoint pathway genes were examined in MC 38 cells treated with *F. nucleatum*. *F. nucleatum* upregulated the expression of G2M checkpoint pathway genes and the cell proliferation of MC 38 cells. *F. nucleatum* facilitated cancer’s use of G2M checkpoint pathways and *F. nucleatum* could be a therapeutic target in Korean GI tract cancer.

## 1. Introduction

Over the past two decades, there has been outstanding progress in the identification of the microbiome that broadly affects the human body. The human body hosts a variety of microbes, and the influences of the gut microbiome on cancer and cancer immunity are well-known. These microbes have been detected even in tumor tissues and can immensely affect tumors [1]. With the development of high-throughput sequencing, also known as next-generation sequencing (NGS), such as 16 s rRNA and whole-transcriptome sequencing (WTS), cancer microbiome research has been actively conducted [2]. The 16 s rRNA sequencing method is a widely used NGS platform upon which to identify taxonomic composition in cancer research. Moreover, the WTS of tumor tissues shows an association between genes and the microbiome. The microbiome is closely related to the mechanisms, genes, and pathways that cancer cells utilize to proliferate [3].

Recent studies have demonstrated that bacteria, such as *Helicobacter pylori*, *Escherichia coli*, *Bacteroides fragilis*, *Salmonella enterica,* and *F. nucleatum* are key players in cancer [4]. It is estimated that approximately half of the global population suffers from an *H. pylori* infection. Through immune response and inflammation, *H. pylori* contributes to the development of gastric adenocarcinoma [5]. By causing inflammation and oxidative stress, *E. coli* strains promote the development of colorectal cancer (CRC). Additionally, genotoxin secreted from *E. coli* is known to generate DNA damage in eukaryotic epithelial cells [6]. *B. fragilis* is common in the entire colon, and enterotoxigenic *B. fragilis* (ETBF) produces metalloprotease *Bacteroides fragilis* toxin (BFT), leading to inflammation and tissue damage in CRC, thereby promoting colon tumorigenesis [7]. *S. enterica* infection causes the activation of the MAPK and AKT pathways, which provoke cellular transformation related to gallbladder cancer [8]. *F. nucleatum* is a Gram-negative obligate anaerobe bacterium and is usually acknowledged as a CRC-promoting bacterium [9]. Cancer is promoted by *F. nucleatum* through the enhancement of host responses that initiate and promote tumors and by encouraging tumor invasion and metastasis [10]. To examine whether the microbiome is associated with gastrointestinal (GI) tract cancer, we looked for a common microbiome among the GI tract cancers and sought to understand how the microbiome is associated with cancer. Additionally, Korean patients were selected for ethnic consistency, which might cause a discrepancy in microbial abundance among different ethnic groups.

After the recognition of the bacterial effect on cancer, there have been many attempts to target the microbiome for cancer therapy [11]. There are a few studies that target the common microbiome in Korean GI tract cancers. Here, *F. nucleatum* was identified as a highly expressed bacteria in Korean GI tract tumor tissues and possible pathways that *F. nucleatum* use to promote cancer. In this study, through WTS analysis by in vitro assay, *F. nucleatum* was demonstrated to encourage cancer cells to proliferate through the G2M checkpoint pathway. Understanding the *F. nucleatum* mechanism and its effect on cancer cells could lead to the discovery of potential therapeutic targets for GI tract cancers.

## 2. Materials and Methods

### 2.1. Public Whole Transcriptome Dataset Analysis

Gastric, esophageal, and CRC WTS data used in this study were obtained from accession numbers GSE113255, GSE130078, and GSE180440 in the Gene Expression Omnibus (GEO) repository [12,13,14]. Quality control and adapter trimming were processed with a fastp pipeline [15]. After filtering the data, HISAT2 was used to align the filtered data to the human database (GRCh38) [16]. HISAT2-aligned bam files were separated into mapped and unmapped bam files with the SAMtools “view” and “sort” commands. Unmapped bam files were converted into fastq formats with the SAMtools “bam2fastq” command [17]. Unmapped fastq files were processed with Kraken2 and the minikraken database to assign taxonomic labels to metagenomic sequences [18]. To quantify reads by genomic features, we used featureCounts [19]. The “DESeq2” package in R version 4.1.2 (Vienna, Austria) was used to normalize the genomic and metagenomic data and extract DEGs and microbiome differential abundance [20]. We used the GSEA platform version 4.1.0 with the MSigDB hallmark gene set for GSEA [21]. The “pheatmap” package in R was used to draw a heatmap plot with z-score-transformed expression data [22]. The “VennDiagram” package in R was used to find overlapped microbiomes in GI tract cancer [23]. The “ggplot2” package in R was used to draw volcano plots, barplots, boxplots, and PCoA plots [24].

### 2.2. Bacterial Culture Growth and Supernatants

We inoculated *F. nucleatum* subsp. *nucleatum* (KCTC 2640) on Reinforced Clostridial Medium (RCM) culture plates and grew it in anaerobic jars in 37 °C incubators for 3 d. We picked isolated strains from the culture plate and grew them in an anaerobic liquid RCM vial in a 37 °C incubator for 2 d. The optical density of *F. nucleatum* supernatants was adjusted to 1.0 at 600 nm in 1 mL RCM broth. The supernatants were obtained at 6000× *g* for 15 min at 4 °C and stored at 4 °C.

### 2.3. Cell Culture Growth, siRNA Transfection, and Treatment of Bacterial Supernatants

The MC 38 cell line was purchased from Kerafast. MC 38 cells were cultured in Dulbecco’s Modified Eagle’s Medium (DMEM) with 10% heat-inactivated fetal bovine serum (FBS) and antibiotic–antimycotic solution (100 units/mL penicillin, 100 μg/mL streptomycin, and 0.25 μg/mL amphotericin B) (Gibco, Life Technologies Co. Ltd., Waltham, MA, USA) in a 37 °C/5% CO_2_ incubator. MC 38 cells (1 × 10^5^) were seeded in 6-well plates with 2.5 mL of DMEM without antibiotics and incubated in a 37 °C/5% CO_2_ incubator until they reached 60% confluency. LipofectamineTM RNAiMAX (Invitrogen, Carlsbad, CA, USA) was used to transfect siRNA oligos. siRNA for mouse Chek1 mRNA were synthesized commercially at Bioneer: siChek1 (forward: 5′-UCUCAAGUCAUGAUUGCUU-3′; reverse: 5′-AAGCAAUCAUGACUUGAGA-3′). The siRNA duplex and lipofectamine mixture were treated in each well for 24 h in a 37 °C/5% CO_2_ incubator. The supernatants of *F. nucleatum* were treated after discarding the siRNA-treated DMEM and incubated for 36 h.

### 2.4. Total RNA Extraction and qRT-PCR Analysis

Total RNA was extracted from MC 38 cells using an RNeasy mini kit (Hilden, Germany, Qiagen) according to the manufacturer’s instructions. cDNA was synthesized using TOPscript RT DryMIX dT18plus (Daejeon, Korea, Enzynomics) according to the manufacturer’s instructions. Gene expression was measured using qRT-PCR, and expression data were handled using StepOne PlusTM software (Waltham, MA, USA, Applied Biosystems). qRT-PCR amplification was achieved using TOPrealTM SYBR Green qPCR PreMIX (Enzynomics). Primers were synthesized by Macrogen: Cdk1 (forward: 5′-AAGGTACTTACGGTGTGGTG-3′; reverse: 5′-CAGGTACTTCTTGAGGTCCA-3′), Cdc25a (forward: 5′-TGGACCTGTCTCCTACACTC-3′; reverse: 5′-GCTCAGTGAGAGCAGCTAAC-3ʹ), Cdc25c (forward: 5ʹ-CTACAGGACCTATCCCACCT-3′; reverse: 5′-CTCTCCACTGCTAAGATTCG-3′), Chek1 (forward: 5′-GGAGTAAGGAAATGCAGGAG-3′; reverse: 5′-GGAGAGTTAAGTGGGTGACA-3′), and Ccnb1 (forward: 5′-GCACCTGGCTAAGAATGTAG-3′; reverse: 5′-GAGCAAGTAAACACGGTAGG-3′). Mouse beta-actin was used as an internal control.

### 2.5. WST-1 Assay

We incubated 100 μL of (5 × 10^3^ cells) MC 38 cells per well in a 37 °C/5% CO_2_ incubator for 24 h. We transfected Chek1 siRNA for 6 h and treated the supernatants of *F. nucleatum* overnight. We treated Quanti-MaxTM (Seoul, Korea, BIOMAX) with 10 μL for 2 h and measured the absorbance at 450 nm absorbance.

### 2.6. Statistical Analysis

All data were tested for normality, and datasets were analyzed using one-way ANOVAs. Post-hoc analysis was conducted using the Bonferroni test. All data results are presented as the mean ± SD. All statistical analyses were performed using GraphPad Prism 9 (La Jolla, CA, USA) and R-4.2.1 (Vienna, Austria) for Mac OS. To compare the two groups (cancer and normal), we used the Wilcoxon test. Statistical significance was set at *p* < 0.05.

## 3. Results and Discussion

To investigate the common bacteria in GI tract cancer, the microbial composition of cancer tissues was compared with those of normal tissues. Open datasets from studies by Kim et al., You et al., and Park et al. in the GEO database were used for identifying the genomic and metagenomic features of GI tract cancer [12,13,14]. An in vitro assay was performed to validate the results of the WTS analysis (Figure 1).

### 3.1. Fusobacterium nucleatum Is Enriched in Gastrointestinal Tract Cancer Tissues

The GI tract is affected by many bacteria. *F. nucleatum* is a well-known bacteria that promotes colorectal carcinogenesis. A recent study showed that it could originate from the oral cavity through the circulatory system [25]. Additionally, there is evidence that *F. nucleatum* plays a role in oral, esophageal, gastric, head and neck, breast, and pancreatic cancers [25,26,27,28]. *F. nucleatum* plays a role in carcinogenesis by inducing a host response for tumor initiation and promotion, and encouraging tumor invasion and metastasis [10]. Humans are highly susceptible to *H. pylori* infection, with approximately half of the global population suffering from this infection. *H. pylori* is related to gastric adenocarcinoma through immune response and inflammation mechanisms. However, *H. pylori* is associated with a decreased risk of esophageal adenocarcinoma [5,29]. Based on previous studies regarding the effects of bacteria on the GI tract, we examined common bacteria as therapeutic targets in GI tract cancer. Korean GI tract cancer has not yet been studied, and there are applicable cohorts in which to examine common bacteria in GI tract tumor tissues. The CRC dataset consisted of 190 WTS data points for 145 tumor tissues and 45 normal colon tissues. The WTS data for esophageal cancer were composed of 23 cancerous tissues and 23 normal adjacent tissues. There were 130 gastric cancer tissues and 10 normal intestinal mucosae in gastric cancer WTS data.

Using a microbiome abundance analysis, it is shown that the expression of *Staphylococcus haemolyticus*, *Micrococcus luteus*, and *Rothia mucilaginosa* in normal colorectal tissues was higher, whereas the expression of *Ralstonia insidiosa*, *R. mannitolilytica*, *F. necrophorum*, and *F. nucleatum* was higher in CRC tissues. In esophageal cancer tissues, *F. nucleatum*, *F. hwasookii*, *Prevotella oris*, and *Leptotrichia trevisanii* were highly expressed. Furthermore, in gastric cancer tissues, *Selenomonas sputigena*, *P. oris*, and *F. nucleatum* were more highly expressed (Figure 2a). *F. nucleatum* is a commonly highly expressed bacteria in all three types of cancer tissues. *P. oris* is a bacteria that is known to originate from the oral cavity and is enriched in gastric cancer samples. Our research showed it was enriched in the ESCC and GC cancer groups [30]. To determine any differences between the relationship of *F. nucleatum* abundance in cancer and normal tissues, the expression count was normalized using DESeq2, which indicated a significant difference (CRC LogFC = 1.755, Adj. *p* value = 0.037; ESCC LogFc = 4.880, Adj. *p* value = 2.36 × 10^11^; GC LogFC = 5.358, Adj. *p* value = 0.009). To specify whether the batch effect of each cohort contributed to this outcome, a principal coordinate analysis (PCoA) was performed with Bray-Curtis distances. The PCoA plot showed that there were few community dissimilarities (Figure 2b). There were six commonly highly expressed bacteria in GI tract cancer. In particular, *F. nucleatum* was the only bacteria that was defined at the species level (Figure 2c, Appendix A). With the microbiome overlap Venn Diagram, it was found that the gastric cancer cohort shared more than 60% of the enriched microbiome with other cohorts. This is thought to be a result of the stomach being a passageway between the esophagus and colon.

### 3.2. G2M Checkpoint Pathway Is Associated with F. nucleatum Abundance in GI Tract Cancer

To determine the possible mechanism of how *F. nucleatum* works, each dataset is divided into two groups, the *F. nucleatum*-high and *F. nucleatum*-low groups, using median values. Using the DESeq2 normalized gene count, Gene Set Enrichment Analysis (GSEA) was performed. Through GSEA analysis, possible pathways with a normalized *p*-value < 0.05 was selected (Figure 3a). We examined whether hallmark genes of the G2M checkpoint were commonly upregulated in all cancer types in the *F. nucleatum* high group (Figure 3b, CRC NES = 1.692, Nom *p*-value = 0.042, FDR q-value = 0.024; ESCC NES = 1.754, Nom *p*-value = 0.008, FDR q-value = 0.039; GC NES = 1.630, Nom *p*-value = 0.039, FDR q-value = 0.452). Major anti-cancer therapies target DNA or cell-division mechanisms. These therapies elicit the activation of cell-cycle checkpoints. The G2M checkpoint pathway is used by cancer cells to avoid apoptosis [31]. Then, the expression of genes related to the G2M checkpoint pathways in cancer and normal tissues was compared. Higher expression of G2M checkpoint pathway genes (*CHEK1*, *CDK1*, *CDC25A*, *CDC25B*, *CDC25C*, *CCNB1*) was observed in cancer tissues, with all comparisons having a *p*-value less than 0.01 (Figure 3c). *CHEK1* and its downstream genes appeared to be related to a common cancerous pathway, and *CHEK1* is a key factor in the checkpoint control of the G2M checkpoint pathway [32]. *CHEK1* is known as a tumor repressor, but recent studies have demonstrated that *CHEK1* can promote cancer cell proliferation. Furthermore, the inhibition of *CHEK1* can be a therapeutic target for cancer by sensitizing cells to DNA-damaging agents, such as ionizing radiation (IR) and antimetabolites, or when used as single agents [32,33,34]. The suppression of *CHEK1* can downregulate *CCNB1*, and this downregulation can impair CRC proliferation. The repression of *CCNB1* can invoke G2M phase arrest and interfere with the expression of *CDC25c* and *CDK1* [35]. Furthermore, genes related to angiogenesis and epithelial–mesenchymal transition (EMT) were investigated, and meaningful significance was found in the GSEA analysis of CRC and ESCC cohorts (Appendix A, CRC angiogenesis NES = 1.641, Nom *p*-value = 0.019, FDR q-value = 0.038; CRC EMT NES = 1.46, Nom *p*-value = 0.128, FDR q-value = 0.101; ESCC angiogenesis NES = 1.780, Nom *p*-value = 0.008, FDR q-value = 0.035; ESCC EMT NES = 1.795, Nom *p*-value = 0.004, FDR q-value = 0.038). Additionally, angiogenesis and EMT are pathways that contribute to cancer proliferation [36]. However, angiogenesis- and EMT-related genes were not always highly expressed in the cancer group, such as *VIM* and *VEGFA* (Figure 3d). To determine if the abundance of *F. nucleatum* affects the expression of the G2M pathway, a correlation analysis was performed between the expression of *F. nucleatum* and G2M pathway-related genes. A weak positive correlation was identified between *F. nucleatum* abundance and the expression of G2M pathway genes (Appendix A).

### 3.3. F. nucleatum Promotes Cancer Cell Proliferation through the G2M Checkpoint Pathway

MC 38 cells, derived from murine colon adenocarcinoma cells, were used to verify the WTS results by in vitro assay, especially for CRC patients [37]. To inhibit the cell proliferation of MC 38 cells, *Chek1* siRNA was treated. We expected that *F. nucleatum* might counteract the activity of *Chek1* siRNA. *F. nucleatum* subsp. *nucleatum*, which is well-known for causing infections in humans, was used [38]. We differed the concentration of supernatants to determine the optimal density. The expression of *Chek1*, *Cdk1*, *Cdc25a*, *Cdc25c*, and *Ccnb1* was examined to confirm the activity of *Chek1* siRNA and the supernatants of *F. nucleatum*. The qRT-PCR results demonstrated that treatment with *F. nucleatum* in a 5% concentration best offset the effects of *Chek1* knockdown. *Chek1* siRNA successfully knocked down the expression of *Chek1* in MC 38 cells, and the supernatant of *F. nucleatum* could not decrease the effect of the knockdown on *Chek1*.

However, the supernatants increased the expression of Cdc25 components, which resulted in the upregulation of *Cdk1* (Figure 4a). In particular, *Cdc25c* is known to be overexpressed in various types of cancer and promotes tumorigenesis. Furthermore, through the interaction between *Cdc25c* and *Cdk1*, cancer cell proliferation can be repressed [35,39]. Through the WST-1 assay, the increased proliferation of MC 38 cells treated with the supernatants of *F. nucleatum* was identified (Figure 4b). These results demonstrated that the knockdown of *Chek1* could inhibit cancer cell proliferation through *Cdc25c* and *Cdk1*-*Ccnb1*, and the treatment of *F. nucleatum* supernatants can overcome this suppression.

In conclusion, GI tract cancers frequently recur, especially GC and CRC, in Koreans [40]. There are known diagnostic biomarkers for GI tract cancers, such as genetic aberrations and molecules [41]. *F. nucleatum* is one of the well-known bacteria that negatively affects the prognosis of cancers by the activation of chemokines that lead to aggressive tumor behavior [25,28]. In our study, it is shown that *F. nucleatum* is enriched in Korean GI tract cancers, and *F. nucleatum* facilitated cancer’s use of the G2M checkpoint pathway. In particular, the expression of *CDC25c* and *CDK1*-*CCNB1* was investigated, which was increased by *F. nucleatum* among diverse paths of the G2M checkpoint. This study suggests that *F. nucleatum* is a potential therapeutic target not only for CRC but also for ESCC and GC in Korean patients. The prevention of *F. nucleatum* from proliferating could be used as a clinical strategy for GI tract cancer patients. Metronidazole, an antibiotic for anaerobic infections, could be used for targeting the broad spectrum of anaerobic organisms, including *F. nucleatum*, in CRC [42].To further investigate how *F. nucleatum* acts in GI tract cancers, metabolites associated with *F. nucleatum,* and the G2M checkpoint pathways, additional examination is needed. Furthermore, additional microbiome studies of the GI tracts of other ethnic populations could elucidate whether enriched *F. nucleatum* in GI tract cancers is the consequence of ethnic attributes.

## Figures and Tables

**Figure 1 microorganisms-10-01896-f001:**
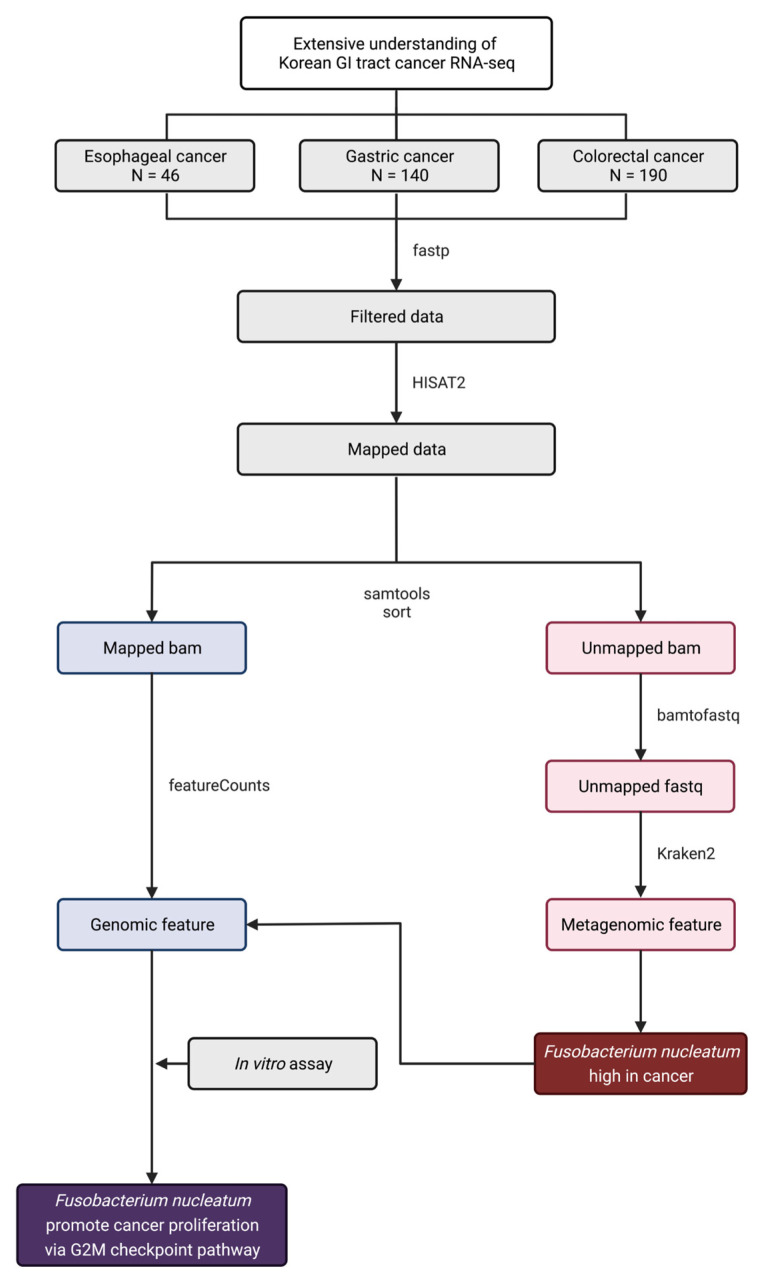
Flow of WTS data processing. Overall flow of WTS data processing for genomic feature and metagenomic feature analysis.

**Figure 2 microorganisms-10-01896-f002:**
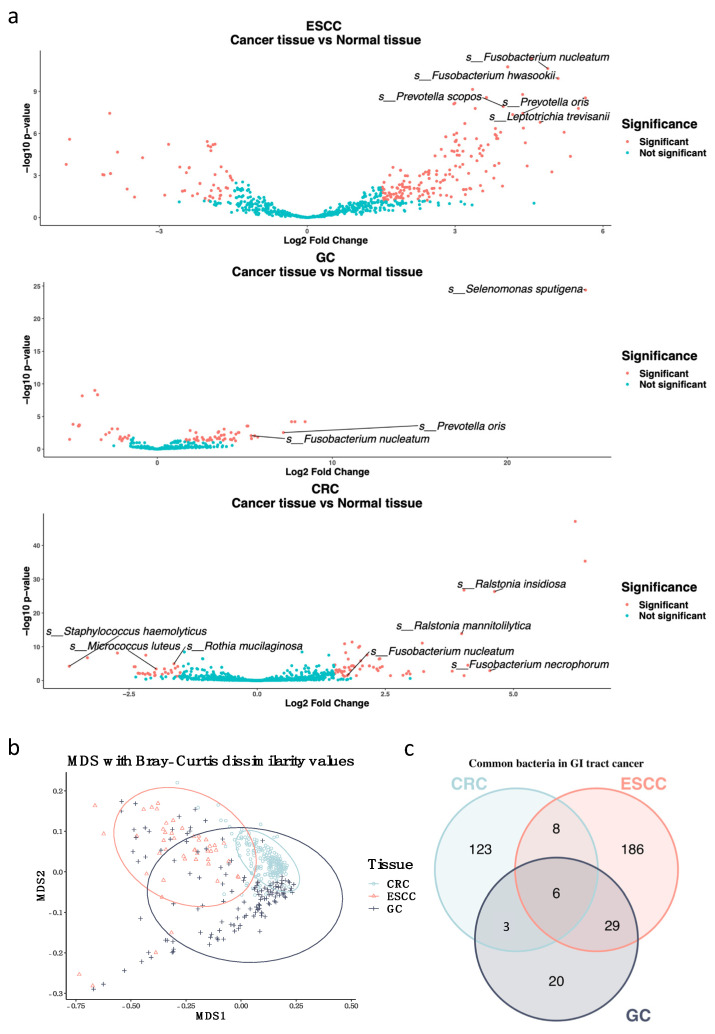
Microbiome differential abundance analysis of GI tract cancer WTS data. (**a**) Volcano plot demonstrating the differential prevalence of the microbiome between cancer and normal tissues from the esophageal (GSE130078), gastric (GSE180440), and colorectal (GSE113255) samples. The microbiome was colored when it surpassed the significance threshold (*p*-value < 0.05, fold change > 1.5). Names of species levels in the microbiome are annotated. (**b**) Principal coordinate analysis (PCoA) plot on the Bray−Curtis dissimilarity indexes between microbiome profiles of different cancer types. (**c**) Venn Diagram showing numbers of overlapping microbiomes between GI tract cancer types.

**Figure 3 microorganisms-10-01896-f003:**
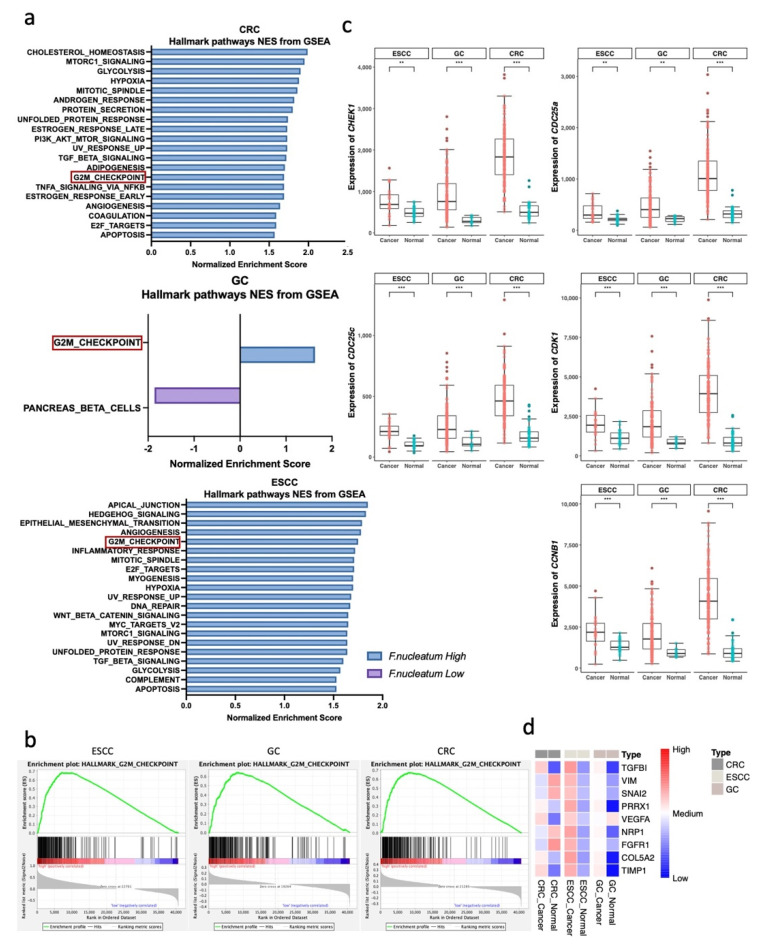
DEGs and GSEA analysis of GI tract cancer WTS data. (**a**) Barplot indicating normalized enrichment score (NES) in *F. nucleatum*−high vs. *F. nucleatum*−low groups for every cancer type. (**b**) Gene set enrichment analysis (GSEA) demonstrating the common pathway (G2M checkpoint) in the *F. nucleatum*−high group of every cancer type (CRC NES = 1.692, Nom *p*-value = 0.042, FDR q-value = 0.024; ESCC NES = 1.754, Nom *p*-value = 0.008, FDR q-value = 0.039; GC NES = 1.630, Nom *p*-value = 0.039, FDR q-value = 0.452). (**c**) Boxplot representing the expression of the G2M pathway−related genes between cancer and normal tissues. ** *p* < 0.01, *** *p* < 0.001. Statistical significance was calculated using the Wilcoxon test. (**d**) Heatmap for the z−score−converted expression of the EMT pathway- and angiogenesis pathway-related genes.

**Figure 4 microorganisms-10-01896-f004:**
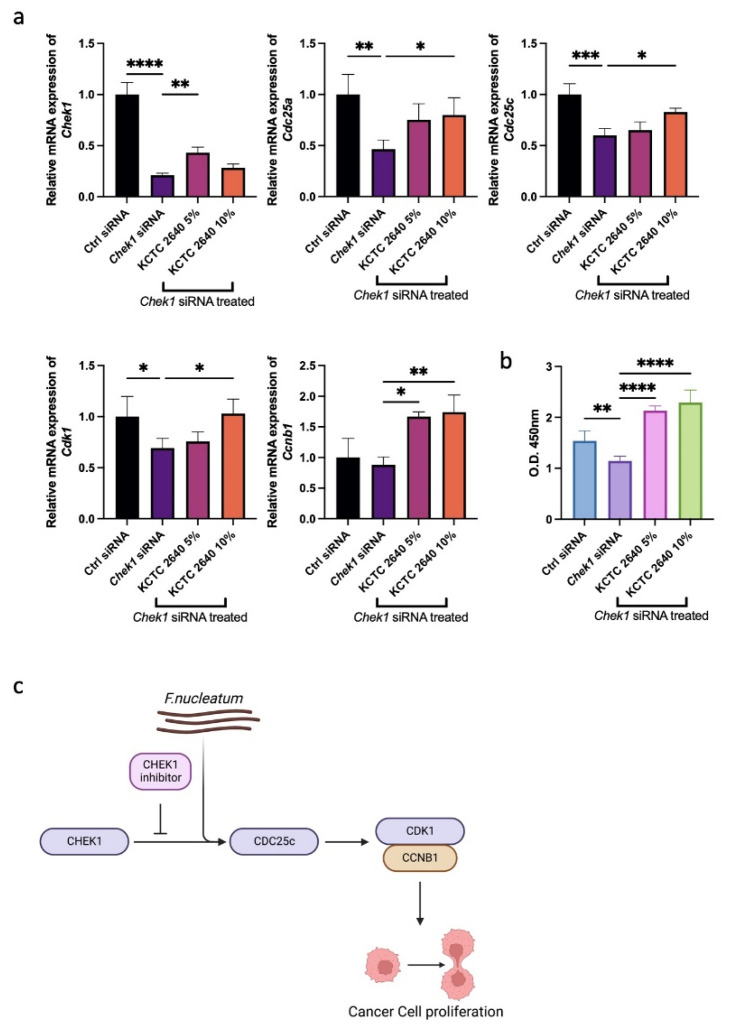
qRT-PCR and WST-1 assay of MC 38 cell lines. (**a**) qRT-PCR assay representing the G2M checkpoint pathway-related genes in MC 38 cell lines treated with *Chek1* siRNA and *F. nucleatum*. (**b**) WST-1 assay demonstrating the cell proliferation of MC 38 cell lines treated with *Chek1* siRNA and *F. nucleatum*. (**c**) Scheme of the G2M checkpoint pathway and effect of *F. nucleatum*. * *p* < 0.05, ** *p* < 0.01, *** *p* < 0.001, **** *p* < 0.0001. Statistical significance was calculated using Bonferroni tests.

## Data Availability

This study used data processed from the Gene Expression Omnibus, accession numbers GSE113255, GSE130078, and GSE180440, for the analysis of gene expression. These data are freely available at https://www.ncbi.nlm.nih.gov/geo/query/acc.cgi?acc=GSE113255 (accessed on 22 February 2022), https://www.ncbi.nlm.nih.gov/geo/query/acc.cgi?acc=GSE130078 (accessed on 9 January 2022), https://www.ncbi.nlm.nih.gov/geo/query/acc.cgi?acc=GSE180440 (accessed on 5 May 2022).

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
