# Peer review of "Whole-Transcriptome Sequencing Reveals Characteristics of Cancer Microbiome in Korean Patients with GI Tract Cancer: Fusobacterium nucleatum as a Therapeutic Target"

_microorganisms, 2022, doi:10.3390/microorganisms10101896_

Round 1

Reviewer 1 Report

The work entitled "Whole transcriptome sequencing reveals characteristics of cancer microbiome in Korean patients with GI tract cancer: Fusobacterium nucleatum as a therapeutic target", clarify some molecular machinery involved in cancer cells, where the authors suggest that F. nucleatum is a potential therapeutic target not only for CRC, but also for ESCC and GC in Korean patients.

The MC 38 cell line was used, however a better explanation about the choice of this cell line should be provided.

The abstract needs better scientific language and text improvement.

Science should be written in the third person and should not be personalised. Phrases which contain "We sought a common ... We selected Korean...", "We tested..." , "we identified...", "In our study...", "we investigated...", "We suggest...", "We..." etc., should be corrected.

The abbreviations must be written out in full the first time they are used. Please, correct this throughout the entire text.

The data results are presented as the mean±SEM, however the correct treatment should be with mean±SD.

The results section needs better organisation. As it is it should be named results and discussion section.

Some molecular mechanisms were clarified in the present work. However, the microbiota of CRC patients is unbalanced, so, to treat this imbalance, pointing to F. nucleatum as a target for treatment does not seem likely for application in cancer patients.

The prevention of F. nucleatum from proliferating seems a strategy more relevant.

The methodology chosen were not innovative for this field of research. Nonetheless, after performing these minor revisions, the paper may be published.

Author Response

Response to Reviewer 1 comments

  • Point 1: The work entitled "Whole transcriptome sequencing reveals characteristics of cancer microbiome in Korean patients with GI tract cancer: Fusobacterium nucleatumas a therapeutic target", clarify some molecular machinery involved in cancer cells, where the authors suggest that  nucleatum is a potential therapeutic target not only for CRC, but also for ESCC and GC in Korean patients. The MC 38 cell line was used, however a better explanation about the choice of this cell line should be provided.
  • Response 1: Thank you for the comments of the reviewer. The authors added explanation and reference about the choice of MC 38 cell line.

Added text : "MC 38 cell, derived from murine colon adenocarcinoma cells, was used to verify the WTS results by in vitro assay, especially for CRC patients."

  • References

Newsome, R.C.; Yang, Y.; Jobin, C. The microbiome, gastrointestinal cancer, and immunotherapy. Journal of Gastroenterology and Hepatology 2022, 37, 263-272.

  • Point 2: The abstract needs better scientific language and text improvement. Science should be written in the third person and should not be personalised. Phrases which contain "We sought a common ... We selected Korean...", "We tested..." , "we identified...", "In our study...", "we investigated...", "We suggest...", "We..." etc., should be corrected. The abbreviations must be written out in full the first time they are used. Please, correct this throughout the entire text.
  • Response 2: The authors modified the personalized terms into scientific language. The authors checked abbreviations throughout the text and added full text at the first-time use.

Orignial text : We sought a common microbiome in GI tract cancers and how the microbiome is related to cancer.

Revised text : The purpose of this study was to determine if there is a common microbiome among GI tract cancers and how the microbiome affects the disease.

Orignial text : We selected Korean patients with GI tract cancer for ethnic consistency

Revised text : To ensure ethnic consistency, Korean patients with GI tract cancer were selected.

Original text : Through GSEA and DEG analyses, we identified a G2M checkpoint pathway upregulated in the F. nucleatum high group.

Revised text : Through Gene Set Enrichment Analysis (GSEA) and Differentially Expressed Genes (DEG) analyses, upregulation of G2M checkpoint pathway was identified in the F. nucleatum high group

Original text : We teseted effects of F. nucleatum on G2M checkpoint pathway genes and cell viability with MC 38 cell lines

Revised text : Cell viability and G2M checkpoint pathway genes were examined in MC 38 cells treated with F. nucleatum.

  • Point 3: The data results are presented as the mean±SEM, however the correct treatment should be with mean±SD.
  • Response 3: The data was presented in mean±SD at first. The authors changed the method from "mean±SEM" to " mean±SD".
  • Point 4: The results section needs better organisation. As it is it should be named results and discussion section.
  • Response 4: The authors changed the name of result section to "Results and Discussion"
  • Point 5: Some molecular mechanisms were clarified in the present work. However, the microbiota of CRC patients is unbalanced, so, to treat this imbalance, pointing to  nucleatumas a target for treatment does not seem likely for application in cancer patients. The prevention of F. nucleatum from proliferating seems a strategy more relevant. The methodology chosen were not innovative for this field of research. Nonetheless, after performing these minor revisions, the paper may be published.
  • Response 5: The authors added text as clinical strategies have not suggested in the article. The authors would like to thank the reviewer for the detailed comments. In a sincere manner, the author answered comments with all words and grammar thoroughly corrected.

Added text : "The prevention of F. nucleatum from proliferating could be used as a clinical strategy for GI tract cancer patients. Metronidazole, an antibiotic for anaerobic infections, could be used for targeting broad spectrum of anaerobic organisms including F. nucleatum in CRC "

Reviewer 2 Report

Some points should be corrected as

1. The abbreviation should be presented the full name at the first time in this manuscript as line 15 (GSEA and DEG; RCM broth).

2. Italic: F. nucleatum subsp. nucleatum. line 83, 101, 121.

3. 6000g........6,000 xg.

4. Please add the description of p-value in Statistical analysis.

5. Please correct that 3. Results to 3. Results and discussion.

6. Lack of Fig. 1 in the  3. Results and discussion.

7. Line 157: p-value.

8. In Fig. 4: lack of star symbols in the Fig. 4A-B.

Author Response

Response to Reviewer 2 comments

  • Point 1: The abbreviation should be presented the full name at the first time in this manuscript as line 15 (GSEA and DEG; RCM broth).
  • Point 2: Italic: nucleatumsubsp. nucleatum. line 83, 101, 121.
  • Point 3: 6000g........6,000 x
  • Point 4: Please add the description of p-value in Statistical analysis.
  • Point 5: Please correct that 3. Results to 3. Results and discussion.
  • Point 6: Lack of Fig. 1 in the  3. Results and discussion.
  • Point 7: Line 157: p-value.
  • Point 8: In Fig. 4: lack of star symbols in the Fig. 4A-B.
  • Response 1-8: The authors modified the text and figure according to reviewer's comments. The authors would like to thank the reviewer for the detailed comments. In a sincere manner, the author answered comments with all words and grammar thoroughly corrected.
